# Priming with Porcine Blood Polypeptide Enhances Salt Tolerance in Wheat Seedlings

**DOI:** 10.3390/plants14192968

**Published:** 2025-09-25

**Authors:** Yong Shen, Yanling Ma, Yiming Yuan, Meitian Dong, Yanan Wang, Jilong Zhou, Jinpeng Yang, Yang Guo, Weiwei Guo, Huifang Wang, Yumei Zhang, Ximei Li

**Affiliations:** 1College of Agronomy, Qingdao Agricultural University/Shandong Engineering Research Center of Germplasm Innovation and Utilization of Salt-Tolerant Crops/Shandong Provincial Key Laboratory of Plant Stress Biology and Genetic Improvement, Qingdao 266109, China; dxdshy@163.com (Y.S.); ma18536496122@163.com (Y.M.); yymydytj@126.com (Y.Y.); tt1912385547@163.com (M.D.); wyn13687631077@163.com (Y.W.); 17852422287@163.com (J.Z.); 13643244099@163.com (J.Y.); 18748434185@163.com (Y.G.); guowei0509@126.com (W.G.); huifangwang2017@163.com (H.W.); 2Academy of Dongying Efficient Agricultural Technology and Industry on Saline and Alkaline Land in Collaboration with Qingdao Agricultural University, Dongying 257000, China

**Keywords:** wheat, salt stress, porcine blood polypeptide, antioxidant enzymes, osmotic regulatory substances

## Abstract

Porcine blood polypeptide (PBP) has been reported to play roles in plant growth. However, its functions in alleviating salt stress in wheat remain unclear. The present study was conducted to investigate the physiological and biochemical mechanisms underlying the effects of PBP on wheat salt tolerance. Morphological analysis showed that PBP-primed seedlings exhibited improved growth performance, significantly greater biomass accumulation, and enhanced root system development. Physiological assessments showed that primed seedlings displayed higher values of *Pn*, *Gs*, *Tr*, *Fv*/*Fm*, *Fv*′/*Fm*′, *Φ*_PSII_, and NPQ, along with increased contents of total chlorophyll, Pro, TSS, and RWC. In addition, the activities of antioxidant enzymes, including SOD, CAT, POD, and APX, were significantly elevated, whereas the levels of H_2_O_2_, O_2_^−^, MDA, and REC were significantly reduced. PCA indicated that antioxidant enzyme activity, osmotic regulation, and ROS accumulation were the major factors associated with the PBP-mediated salt stress response. Furthermore, qRT-PCR analysis suggested that exogenous PBP might enhance wheat salt tolerance by coordinately modulating multiple molecular mechanisms. Taken together, this study broadens the potential applications of PBP by demonstrating its capacity to improve wheat salt tolerance.

## 1. Introduction

Soil salinization is a worldwide ecological environmental problem. The current global saline land is about 954 million hectares, meaning about 20% of the cultivated land area has been affected by salt damage to varying degrees [1]. Salt stress is one of the main reasons causing global crop yield reduction [2,3]. Wheat, one of the most important food crops widely distributed all over the world [4], is also suffering serious damage from salt stress.

Previous studies hold that salt stress first induces osmotic stress, then ion toxicity and oxidative stresses follow [5,6,7]. Osmotic stress often causes physiological drought [8]. High concentration of Na^+^ accumulated in cells can cause cell damage and interfere with K^+^ absorption/metabolism [6,9]. Excessive accumulation of reactive oxygen species (ROS) can lead to lipid peroxidation of cell membrane [10]. All of the above will lead to difficulty in water absorption, changes in membrane permeability, disorder of physiological and biochemical metabolism, decline of photosynthetic performance, stagnation of growth and development, and so on. Ultimately, the end result is a significant reduction in wheat yield [11,12].

Plants have evolved various mechanisms to counter these adverse effects. Some halophytes have evolved special structures such as salt glands to adapt to salt stress [13]. Generally, when exposed to salt stress, plants will transfer absorbed salt ions from cytoplasm to vacuole through transmembrane proteins to achieve ion compartmentalization [14]. In addition, the plant antioxidant system could be activated to clear the large number of ROS, thus bringing it to a new homeostasis level. Antioxidant enzymes, such as superoxide dismutase (SOD), catalase (CAT), peroxidase (POD), and ascorbate peroxidase (APX), play important roles in this process [15]. Osmotic regulation, one of the most basic mechanisms of plant stress resistance, also plays a crucial role in salt stress tolerance. In the early stage, inorganic ions such as K^+^ and Cl^−^ play a major role [16]. With the prolongation of stress time, organic osmotic regulatory substances gradually dominate, mainly including alcohols classes, amino acids and their derivatives, as well as sugars and their derivatives [17]. In addition, levels of plant endogenous hormone, such as abscisic acid (ABA) and jasmonic acid (JA), also change to varying degrees [18].

Up to now, more and more studies have shown that exogenous plant growth regulators can improve wheat salt tolerance through the aforementioned pathways. For example, exogenous JA reduced ROS accumulation by improving the activity of antioxidant enzymes and the concentration of antioxidant compounds [19]; exogenous melatonin (MLT) regulated polyamine metabolism [20]; exogenous salicylic acid (SA) increased the contents of photosynthetic pigment and osmotic regulatory substances, as well as activities of root and antioxidant enzymes [21]; exogenous ABA at an appropriate concentration adjusted the levels of ion and organic solute [22]; and Z-3-HAC regulated the antioxidant and osmoregulatory systems [23]. In order to enhance the production of wheat in saline-alkali soil, new highly effective plant growth regulators should be further explored.

Polypeptides, with more than 2 and less than 100 peptide bonds, are identified as another new type of plant growth regulators discovered after auxin (IAA), gibberellin (GA), ethylene (ETH), cytokinin (CTK), and ABA [24], showing an extremely important role in regulating plant growth, development, reproduction, and response to biological/abiological stresses [25,26,27,28]. Porcine blood, with abundant sources, is a high-quality resource for polypeptide production. Previous studies have shown that porcine blood polypeptide (PBP) played roles in promoting seed germination and seedling growth of Chinese cabbage under salt stress [29], improving yield and quality of wheat [27], and promoting the growth, development, and nutrient absorption of sweet potato seedlings under low-temperature stress [30]. In addition, Zhou et al. [28] found that exogenous pig blood-derived protein hydrolysate effectively enhanced tomato salt tolerance by regulating osmotic regulatory substance biosynthesis, ion homeostasis, and antioxidant system responses.

Generally speaking, there are still very limited reports on the roles of PBP in plant growth and stress resistance. In view of the abundant source of PBP and severe salt damage to wheat production, PBP and a salt-tolerant wheat variety were applied as exogenous regulator and plant material, respectively, in the present study to research the effects of exogenous PBP on wheat seedlings under salt stress and its physiological and biochemical mechanism through a liquid culture experiment.

## 2. Results

### 2.1. Exogenous PBP Promoted the Growth and Dry Matter Accumulation of Wheat Seedlings Under Salt Stress

As visually shown in Figure 1A,B, exogenous application of PBP enhanced wheat seedling growth under both normal and salt stress conditions. Salt stress affected wheat seedling growth negatively, as indicated by the significant decreases in maximum root length (MRL), shoot fresh weight (SFW), root fresh weight (RFW), and root dry weight (RDW) (Figure 1). Priming of PBP significantly alleviated the adverse effects of salt stress, as shown by the fact that seedling height (SH, 9.07%), SFW (21.54%), RFW (22.61%), and RDW (17.77%) of seedlings primed with PBP were significantly higher than those un-primed under salt stress conditions, respectively (Figure 1).

As is well-known, the root system is of great significance for plant growth and development, so root morphology was investigated in detail in this study. The roots of seedlings primed with PBP showed distinct advantages regardless of normal or salt stress conditions (Figure 2A). Under salt stress conditions, the total root length (TRL), total root surface area (TRSA), and total root volume (TRV) were significantly decreased, while average root diameter (ARD) was significantly increased (Figure 2B–E). Whereas the application of PBP before salt stress significantly increased TRL, TRSA, and TRV by 27.37%, 30.36%, and 60.86%, respectively, compared to the salt stressed control (Figure 2B–D). However, ARD significantly decreased (9.37%, Figure 2E).

### 2.2. Exogenous PBP Alleviated the Inhibition of Salt Stress on Photosynthesis in Wheat Seedlings

As shown in Figure 3, under normal conditions, exogenous PBP significantly increased net photosynthetic rate (*Pn*) and stomatal conductance (*Gs*). Salt stress dramatically decreased *Pn*, *Gs*, and transpiration rate (*Tr*), whereas exogenous PBP significantly reduced the adverse effects of salt stress on seedlings. Specifically, *Pn*, *Gs*, and *Tr* were significantly increased by 92.76%, 31.96%, and 27.29%, respectively. Notably, there was no significant change in intercellular CO_2_ concentration (*Ci*) among the four treatments.

Under normal conditions, exogenous PBP significantly increased the maximal photochemical efficiency of photosystem II (PSII) (*Fv*/*Fm*), the photochemical activity of PSII (*Fv*′/*Fm*′), and chlorophyll content (Cc), while no effect on the quantum yield of PSII (*Φ*_PSII_) and the nonphotochemical quenching (NPQ) (Figure 4). All these parameters were significantly decreased by salt stress. Notably, the application of PBP before salt stress significantly alleviated the adverse effects of salt stress. Under salt stress conditions, the above-mentioned parameters of seedlings primed with PBP were significantly higher than those un-primed by 42.75%, 46.76%, 5.07%, 35.32%, and 41.68%, one after another (Figure 4).

### 2.3. Exogenous PBP Reduced ROS Accumulation and Alleviated Membrane Damage to Wheat Seedlings Under Salt Stress

To determine whether exogenous PBP has an impact on ROS accumulation, the accumulation of H_2_O_2_ and O_2_^−^ was analyzed by diaminobenzidine (DAB) and nitro blue tetrazolium chloride monohydrate (NBT) histochemical staining assays, respectively, as well as quantitative analysis (Figure 5). The results showed that, under normal conditions, exogenous PBP had almost no effect on H_2_O_2_ accumulation and significantly decreased O_2_^−^ accumulation. Salt stress dramatically increased H_2_O_2_ and O_2_^−^ content by 30.02% and 150.05%, respectively, while they were dramatically decreased by exogenous PBP by 8.26% and 14.06%, respectively (Figure 5C,D). All these results could also be clearly observed from the qualitative analysis based on histochemical staining assays (Figure 5A,B). In addition, content of malonaldehyde (MDA) and relative electrolyte conductivity (REC) were both significantly increased by salt stress, while exogenous PBP significantly decreased them by 6.27% and 41.42%, respectively (Figure 5E,F).

### 2.4. Exogenous PBP Enhanced the Activities of Antioxidant Enzymes and Increased the Contents of Osmotic Regulatory Substances in Wheat Seedlings Under Salt Stress

Under normal conditions, there was almost no change in activities of SOD and CAT caused by exogenous PBP, whereas a significant decrease in POD activity was observed while a significant increase in APX activity occurred (Figure 6). Under salt stress conditions, the activities of the above antioxidant enzymes were all significantly increased: SOD by 18.52%, CAT by 18.61%, POD by 24.97%, and APX by 77.05% (Figure 6). Moreover, exogenous PBP further significantly increased the activities of SOD by 10.10%, CAT by 8.73%, POD by 4.53%, and APX by 19.48% (Figure 6).

Under normal conditions, there was almost no change in the concentration of proline (Pro) caused by exogenous PBP, whereas significant increases in concentrations of total soluble sugar (TSS) and free amino acids (FAA) were observed (Figure 7A–C). Under salt stress conditions, the concentrations of the above osmotic substances were all significantly increased: Pro by 1193.92%, TSS by 27.94%, and FAA by 188.13% (Figure 7A–C). Again, exogenous PBP further significantly increased the concentrations of Pro by 25.60% (Figure 7A) and TSS by 13.40% (Figure 7B). However, a significant decrease of 28.02% in FAA concentration was observed (Figure 7C). In addition, priming of PBP under salt stress conditions significantly increased leaf relative water content (RWC) by 23.57%, effectively counteracting the significant reduction caused by salt stress (Figure 7D).

### 2.5. Principal Component Analysis

In order to discover the main factors that push wheat seedlings to respond to salt stress, a principal component analysis (PCA) of the four treatments was performed in this study. The two components of PCA collectively explained 77.6% of data variability (Figure 8). The first PC (PC1) accounted for 65.5% of the total qualitative variation and had POD, Pro, FAA, H_2_O_2_, O_2_^−^, MDA, and REC with high negative loadings (Figure 8B). The second PC (PC2) accounted for 12.1% of the total qualitative variation and had SOD, CAT, APX, TSS, *Fv*/*Fm*, and *Fv*′/*Fm*′ with high positive loadings (Figure 8B). SOD, CAT, POD, APX, Pro, TSS, FAA, H_2_O_2_, O_2_^−^, and MDA were located toward the negative end of the PC1 axis and the positive end of the PC2 axis in the second quadrant (Figure 8B). It corresponded with the result of “PBP + NaCl” (Figure 8A). In conclusion, the antioxidant enzymes, osmotic regulatory substances, and ROS accumulation were the most important factors in response to PBP under salt stress.

### 2.6. The Influence of Exogenous PBP on the Expression of Stress Response Genes in Wheat Seedlings Under Salt Stress

As shown in Figure 9, which presents the qRT-PCR results of six classical stress-responsive genes, the overall trend was that salt stress induced an upregulated expression of these genes, and exogenous PBP under salt stress further enhanced this induction. For *TaNHX1*, *TaSOS1*, *TaCIPK*, and *TaP5CS*, gene expression levels in seedlings sprayed with exogenous PBP were significantly higher than those in unsprayed seedlings at almost all time points (Figure 9A–D). For *TaWRKY17*, expression was significantly higher in PBP-treated seedlings after 6, 24, 48, and 96 h of salt stress, whereas the opposite pattern was observed at other time points (Figure 9E). For *TaSOD1*, expression was significantly higher only at 24 h in PBP-treated seedlings, while at 6, 12, 48, and 72 h, expression levels were lower than those in unsprayed seedlings (Figure 9F).

## 3. Discussion

Generally, salt stress usually leads to a significant accumulation of ROS in plant cells, which will lead to membranous peroxidation, damage the integrity of the cell membrane, decrease selectivity for substances, and ultimately pose a threat to the normal growth of plants [7,10]. As is well-known, SOD, CAT, POD, and APX are major components of the ROS scavenging systems under salt stress conditions [15]. Among them, SOD could change harmful O_2_^−^ to the less dangerous H_2_O_2_ [31], while CAT and POD could decompose H_2_O_2_ into H_2_O [32]. APX even directly avoids the formation of H_2_O_2_ by oxidizing ascorbic acid into dehydroascorbic acid and water [33]. In this study, salt stress induced an increase in antioxidant enzyme activities to scavenge ROS, and exogenous PBP under salt stress conditions induced a further significant activity increase (Figure 6), leading to a significant decline in ROS accumulation, indicating that PBP could effectively reduce the excessive accumulation of ROS induced by salt stress (Figure 5A–D). As a result, the content of MDA decreased significantly (Figure 5E), which is an important indicator of membrane lipid peroxidation level [34]. Subsequently, the damage to the cell membrane was reduced, leading to improved selective permeability, as evidenced by a significant decrease in REC that is an indicator of leaf cell membrane stability [35] (Figure 5F).

The aforementioned factors are of vital importance for maintaining normal cellular physiological functions, particularly in ensuring the proper functioning of photosynthesis. Firstly, the antioxidant system helps maintain the structural integrity of thylakoid membranes and prevent oxidative damage to photosynthetic apparatus by inhibiting membrane lipid peroxidation, thereby ensuring the efficient operation of light reactions [36]. Secondly, the key photosynthetic enzymes, such as rubisco, are sensitive to oxidative stress. The antioxidant system could protect the catalytic centers of these enzymes by scavenging ROS, thereby maintaining their activity and ensuring the smooth operation of the Calvin cycle [37]. Thirdly, the antioxidant system contributes to the stability of PSII and PSI by protecting the components of the photosynthetic electron transporter chain from oxidative damage, thereby sustaining the efficiency of light energy conversion [38]. Lastly, the antioxidant system could modulate stomatal movement by regulating cellular H_2_O_2_ levels, thereby maintaining relatively high stomatal conductance and ensuring sufficient CO_2_ supply [39,40], which helps mitigate the decline in *Pn*.

As mentioned previously, the osmotic regulation system also plays a crucial role in salt stress tolerance. Firstly, plants could accumulate osmotic regulatory substances to decrease cellular osmotic pressure, thereby retaining water in leaves and maintaining cell turgor, which is beneficial to sustaining a high *Pn* [41,42]. Secondly, osmotic regulation helps maintain cell turgor pressure, thereby preserving the activity of photosynthesis-related enzymes [43,44]. Thirdly, osmotic regulation contributes to the structural and functional stability of chloroplasts [45], which serve as essential sites for the absorption, transmission, distribution, and conversion of light energy during photosynthesis [46], thereby ensuring sufficient energy supply for both light and dark reactions [43]. Lastly, plants precisely regulate stomatal aperture through osmotic adjustment, which not only minimizes water loss but also maintains high stomatal conductance, thereby ensuring sufficient CO_2_ supply for photosynthesis [47,48].

The results in this study showed that salt stress induced an increase in concentrations of Pro, TSS, and FAA, and exogenous PBP under salt stress conditions induced a further significant increase in concentrations of Pro and TSS, leading to a significant increase in RWC (Figure 7). In particular, Pro, the most important osmotic regulatory substance, showed a drastic increase (1193.92%) under salt stress, and exogenous PBP further increased it by 25.60% (Figure 7A). It has been reported that, beyond its role in osmotic regulation, Pro could prevent cell dehydration under stress conditions due to its hydrophilic property, preserve the conformation and activity of photosynthetic enzymes by maintaining cellular hydration, protect the photosynthetic apparatus from oxidative damage, stabilize membrane permeability, and maintain PSII stability [42]. In addition, TSS, one of the small-molecule organic compounds, increased by 27.94% under salt stress and by an additional increase of 13.40% following exogenous PBP in this study (Figure 7B), could facilitate water absorption and retention, and protect enzymatic activity, thereby indirectly sustaining the normal function of the photosynthetic apparatus [49]. As for FAA, its content was significantly increased under salt stress, whereas exogenous PBP application resulted in a decrease in this study (Figure 7C). This phenomenon could be attributed to the priming effect of PBP, which is rich in amino acids and polypeptides, thereby reducing the need for the accumulation of endogenous free amino acids.

In summary, under salt stress conditions, exogenous PBP significantly enhanced the activities of antioxidant enzymes, thereby reducing ROS accumulation and ultimately alleviating damage to the cell membrane. At the same time, the contents of osmotic regulatory substances were significantly increased, which was beneficial for maintaining the RWC of leaves, reducing membrane permeability, and preserving organelle integrity. Collectively, these physiological adjustments were conducive to the normal operation of photosynthesis, as evidenced by the significant increases in gas exchange parameters (*Pn*, *Gs*, and *Tr*), chlorophyll fluorescence parameters (*Fv*/*Fm*, *Fv*′/*Fm*′, *Φ*_PSII_ & NPQ), and total chlorophyll content in this study (Figure 3 and Figure 4).

Photosynthesis, as a fundamental metabolic pathway, is essential for sustaining crop growth and achieving high yields. The improvement of photosynthetic indicators by PBP priming under salt stress ensured the normal process of photosynthetic carbon assimilation, thereby facilitating biomass accumulation. The results in this study showed that seedlings primed with PBP exhibited better growth and higher biomass accumulation than those un-primed (Figure 1). Better growth induced by exogenous PBP was also reflected in the improvement of root system morphology (Figure 2). A possible explanation is that enhanced photosynthesis increased the accumulation of carbon assimilation products, which were subsequently transported to the roots via phloem sieve tubes, thereby promoting root system development. In return, the improved root system enhanced the absorption of water and nutrients, thereby allowing the leaves to perform photosynthesis more efficiently. Consequently, exogenous PBP alleviated the detrimental effects of salt stress on wheat.

Generally speaking, the present study has relatively comprehensively investigated the underlying physiological and biochemical mechanisms by which PBP enhances salt tolerance in wheat. PBP, a hydrolysis product of porcine blood protein, could be efficiently absorbed when sprayed on leaf surfaces due to its small molecular weight, thereby promoting dry matter synthesis, which might also contribute to the improved salt tolerance of wheat seedlings. Nevertheless, the potential molecular mechanisms underlying these physiological and biochemical responses remain to be further elucidated. It has been reported that endogenous polypeptides frequently initiate intracellular downstream signaling cascades by binding to corresponding receptors, and thereby regulate plant growth, development, and stress resistance [50]. For example, AtCLE9 regulated stomatal closure through the mitogen-activated protein kinase (MAPK) cascade pathway, thereby reducing water loss and enhancing the drought tolerance of *Arabidopsis* [51].

The MAPK signaling pathway is a highly conserved signaling network in eukaryotes and is responsible for transmitting extracellular stimulus signals from cell membrane to nucleus, thereby playing a central regulatory role in various physiological processes, such as cell proliferation, differentiation, and stress responses [52]. The mechanism by which the MAPK signaling pathway contributes to salt stress response can be summarized as follows. Through a three-level kinase cascade of MAPKKK→MAPKK→MAPK, it ultimately regulates diverse downstream effector molecules [53]. The activated MAPKs could translocate into the nucleus and regulate the expression of downstream genes by phosphorylating specific transcription factors [54], a mechanism which is also presumed to be involved in the response to salt stress. In addition, MAPK signaling acts in concert with Ca^2+^-dependent pathways, such as the SOS pathway, to maintain ion homeostasis and thereby enhance salt stress tolerance [55]. Moreover, it participates in the biosynthesis and transport of osmotic regulatory substances, as well as in the regulation of antioxidant enzyme activities, thus contributing to salt stress resistance [56,57].

PBP, mainly composed of amino acids and polypeptides, might function analogously to endogenous polypeptides and activate the MAPK signaling pathway. This hypothesis could be partly supported by the high expression of *TaNHX1*, *TaSOS1*, and *TaP5CS* (Figure 9), together with the enhanced activities of antioxidant enzyme and the increased concentrations of osmotic regulatory substances (Figure 6 and Figure 7). Of course, the above were merely indirect pieces of evidence or speculations. More direct evidence for this hypothesis could be obtained by priming PBP on wheat *mapk* mutants and analyzing their salt tolerance. At least, the present results demonstrated that exogenous PBP induced a higher expression of salt stress-related genes, thereby enhancing wheat salt tolerance. It is widely known that *NHX1* and *SOS1* genes play important roles in Na^+^ compartmentalization and efflux, respectively [55,58]; *CIPK23* regulates K^+^ uptake and contributes to ion homeostasis [59], and P5CS encodes a key enzyme in the Pro biosynthesis pathway, with Pro functioning as an important osmotic regulator under salt stress [60]. In this study, the upregulated expression of *TaNHX1*, *TaSOS1*, *TaCIPK23*, and *TaP5CS* indicated that exogenous PBP might enhance wheat salt tolerance by coordinately modulating multiple molecular mechanisms. Additionally, the fluctuating expression levels of *TaWRKY17* and *TaSOD1* suggested that these two genes might also contribute to the enhancement of wheat salt tolerance induced by PBP. However, their roles appeared to be less pronounced compared with those of the aforementioned genes. Furthermore, the marked increase in SOD enzyme activity induced by PBP indicated that other members of the SOD gene family might be primarily responsible for this effect.

Lastly, a limitation of this study is that the experiments on alleviating salt stress through PBP were conducted only at the seedling stage, although this stage is indeed the most sensitive period. Further experiments should be carried out in saline-alkali soil to test the effect of exogenous PBP under more realistic conditions. Then a more comprehensive evaluation of exogenous PBP in alleviating salt stress in wheat could be achieved by measuring photosynthetic indicators during the flowering period, biomass accumulation at maturity, and final yield together with its constituent factors.

## 4. Materials and Methods

### 4.1. Materials

The drought-resistant and salt-tolerant wheat variety “Qingmai11”, released by Qingdao Agricultural University and approved by Shandong Province (Lushenmai20210018) and the state (Guoshenmai20241053) in 2021 and 2024, respectively, was used in this study. Seeds used in this study were harvested in 2023, and their sterilization and germination were followed to the methods described by Zhang et al. [61]. Then seedling cultivation was performed according to the methods described by Li et al. [62]. Finally, twelve boxes, each containing 48 uniformly sized seedlings of four-leaf periods, were used for subsequent experiments. PBP was purchased from Shanxi Baichuan Biotechnology Co., Ltd. (Xi’an, China). The general extraction and processing procedures are as follows: pre-treatment of pig blood—extraction of pig blood protein—enzymatic hydrolysis—membrane filtration—refinement processing—spray drying. The protein content of the final product is 98%, and the molecular weight distribution range and proportion are as follows: those with a molecular weight less than 189 account for 13.77%, those between 189 and 500 account for 27.08%, those between 500 and 1000 account for 27.25%, those between 1000 and 2000 account for 17.18%, those between 2000 and 3000 account for 8.03%, those between 3000 and 5000 account for 5.28%, those between 5000 and 10,000 account for 1.41%, and those greater than 10,000 account for 0.00%.

### 4.2. Experimental Design

The 12 boxes of seedlings were randomly divided into two batches. One batch underwent foliar spraying of 4 g/L PBP for five consecutive days (once a day), while the other batch was sprayed with the same amount of distilled water. The spray bottles used in laboratory were applied as spraying tools, the droplets produced by which were approximately 150 µm in size. According to our preliminary experiments (five concentrations: 0.5, 1, 2, 4, and 8 g/L PBP were compared, and detailed data were shown in Appendix A), the effect of 4 g/L PBP is the most obvious. After pretreatment for 3 days, half of the seedlings treated with PBP and distilled water were exposed to salt stress of 250 mmol L^−1^ NaCl. Finally, there were four treatments: control, PBP, NaCl, and PBP + NaCl. Five days after salt stress, morphological and physiological parameters were measured with three independent biological replications for each treatment. In addition, regarding NaCl and PBP + NaCl treatments, three leaf replicates from each treatment were collected separately 0 h, 1 h, 6 h, 12 h, 24 h, 48 h, 72 h, and 96 h after the salt stress treatment for RNA extraction.

### 4.3. Measurement of Morphological Indicators

One representative box and one representative seedling for each treatment were selected and photographed, respectively. Then, three randomly chosen seedlings from each box were used as a repeat to measure fresh and dry weight according to the methods of Tian et al. [63]. A total of nine seedlings for each treatment were used to measure seedling height and maximum root length. Root morphology scanning and data analysis were conducted according to the methods described by Li et al. [62]. In addition, one representative picture of root morphology for each treatment was selected and displayed.

### 4.4. Determination of Photosynthetic Indexes

The portable photosynthesis system (Li-COR 6800, Lincoln, NE, USA) was used to determine gas exchange parameters between 9:00 am and 11:00 am. Detailed conditions in the leaf chamber and assay method were the same as those described by Tian et al. [63]. The imaging pulse amplitude modulation (PAM) fluorometer (IMAG-MAXI; Heinz Walz, Effeltrich, Germany) was used to determinate chlorophyll fluorescence according to the methods described by Ahammed et al. [64]. Then, the chlorophyll fluorescence parameters were calculated according to the formulas described by Kramer et al. [65]. Furthermore, the representative *Fv*/*Fm* image from each treatment was exported and displayed. In addition, the ultraviolet–visible spectrophotometer (UV3200, Mapada Instruments Co., Ltd., Shanghai, China) was used to determine the total chlorophyll content, fresh leaf processing, and detailed measurement steps which were carried out as described by Tian et al. [63] and Lichtenthaler & Wellburn [66], respectively.

### 4.5. Measurement of RWC and REC

RWC and REC were measured according to the methods of Jensen et al. [67] and Griffith & McIntyre [68], respectively. Calculations of RWC and REC were conducted as described by Li et al. [62].

### 4.6. Accumulation Analysis of H_2_O_2_, O_2_^−^, and MDA

Histochemical staining detection of H_2_O_2_ and O_2_^−^ were conducted using DAB and NBT, respectively, according to the methods of Tian et al. [63]. Concentration measurement of H_2_O_2_, O_2_^−^, and MDA were conducted using the corresponding kit, respectively, with item numbers of G0112W48, G0116W48, and G0109W in sequence, which were purchased from Suzhou Greast Biotechnology Co., Ltd. (Suzhou, China). The main detection principles are as follows. H_2_O_2_ reacts with titanium salt to form a yellow peroxide–titanium complex precipitate, whose maximum absorption peak occurs at 415 nm after dissolution in concentrated sulfuric acid. O_2_^−^ reacts with hydroxylamine to generate nitrite (NO_2_^−^), which subsequently forms a pink azo dye with maximum absorbance at 540 nm under the action of p-aminobenzenesulfonic acid and α-naphthylamine. Under high-temperature acidic conditions, MDA reacts with thiobarbituric acid (TBA) to form a red adduct with maximum absorbance at 532 nm, then MDA content can be calculated using the differential absorbance (A_532_–A_600_). The specific determination procedures and calculation formulas can be found in the manual.

### 4.7. Activity Analysis of Antioxidant Enzymes

Enzyme activities of SOD, CAT, POD, and APX were detected using the corresponding kit, respectively, with item numbers of G0103W, G0105W, G0107W, and G0203F in sequence, which were also purchased from Suzhou Greast Biotechnology Co., Ltd. The main detection principles are as follows. NBT reacts with O_2_^−^ triggered by xanthine oxidase (XO) to form a colored substances with maximum absorbance at 560 nm, while SOD could eliminate O_2_^−^, thereby inhibiting the formation of colored substances. CAT catalyzes the decomposition of H_2_O_2_ into H_2_O and oxygen, while residual H_2_O_2_ reacts with a novel substrate to form a colored product with maximum absorbance at 510 nm. Under the catalysis of POD, H_2_O_2_ oxidizes guaiacol to produce a reddish-brown product with maximum absorbance at 470 nm. The intermediate complex formed by APX and H_2_O_2_ can oxidize ascorbic acid (AsA), so APX activity could be calculated by measuring the rate of AsA oxidation at 290 nm. The specific determination procedures and calculation formulas were detailed in the manual.

### 4.8. Content Determination of Osmotic Regulatory Substances

Contents of Pro, TSS, and FAA were detected using the corresponding kit, respectively, with item numbers of G0111W48, G0501W48, and G0415W in sequence, which were again purchased from Suzhou Greast Biotechnology Co., Ltd. The main detection principles are as follows. Pro dissolved in sulfosalicylic acid reacts with acidic ninhydrin to form a red chromophore with maximum absorbance at 520 nm. Under the action of concentrated sulfuric acid, soluble sugars produce furfural or hydroxymethyl furfural through dehydration reactions, which then undergo dehydration condensation with anthrone to form derivatives of furfural with maximum absorbance at 620 nm. Under heating acidic conditions, FAA reacts with ninhydrin to form a blue–purple compound with characteristic absorption peak at 570 nm. The specific determination procedures and calculation formulas were detailed in the manual.

### 4.9. RNA Isolation and Expression Analysis of Stress Response Genes

According to the methods described by Li et al. [69], total RNA extraction, concentration measurement, and integrity detection were carried out, as well as first strand cDNA synthesis and qRT-PCR analyses. The primers used in the qRT-PCR analysis were listed in Appendix A, with *TaActin* being used as the internal control.

### 4.10. Statistical Analysis

All data collected were statistically analyzed using one-way ANOVA with the SPSS statistical software package (v22.0, SPSS Inc., Chicago, IL, USA). Duncan’s test (*p* < 0.05) was performed to evaluate differences in each treatment. PCA was conducted using Origin Pro (v2024, Origin Lab Corporation, Northampton, MA, USA).

## 5. Conclusions

Under salt stress conditions, exogenous PBP significantly enhanced the photosynthetic capacity of wheat seedlings by regulating antioxidant and osmoregulatory systems, thereby promoting biomass accumulation. In summary, exogenous PBP effectively alleviated the damage caused by salt stress in wheat seedlings.

## Figures and Tables

**Figure 1 plants-14-02968-f001:**
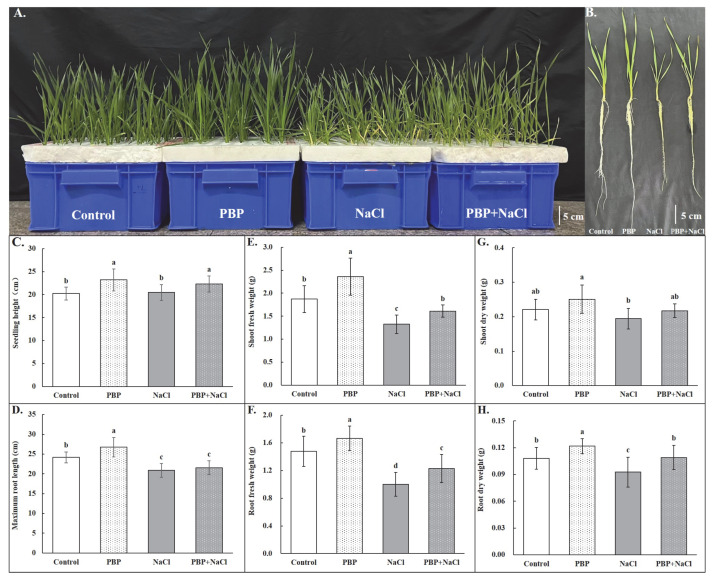
Exogenous PBP application conferred salt tolerance of wheat seedlings. (**A**) The overall growth of wheat seedlings under normal/salt stress conditions with/without PBP priming, (**B**) growth of representative single wheat seedling, (**C**) seedling height (SH), (**D**) maximum root length (MRL), (**E**) shoot fresh weight (SFW), (**F**) root fresh weight (RFW), (**G**) shoot dry weight (SDW), and (**H**) root dry weight (RDW). Bars are the standard deviations (SDs) of three independent replicates (*n* = 3). Error bars labeled with different letters indicate significant differences at *p* < 0.05 between treatments according to Duncan’s test.

**Figure 2 plants-14-02968-f002:**
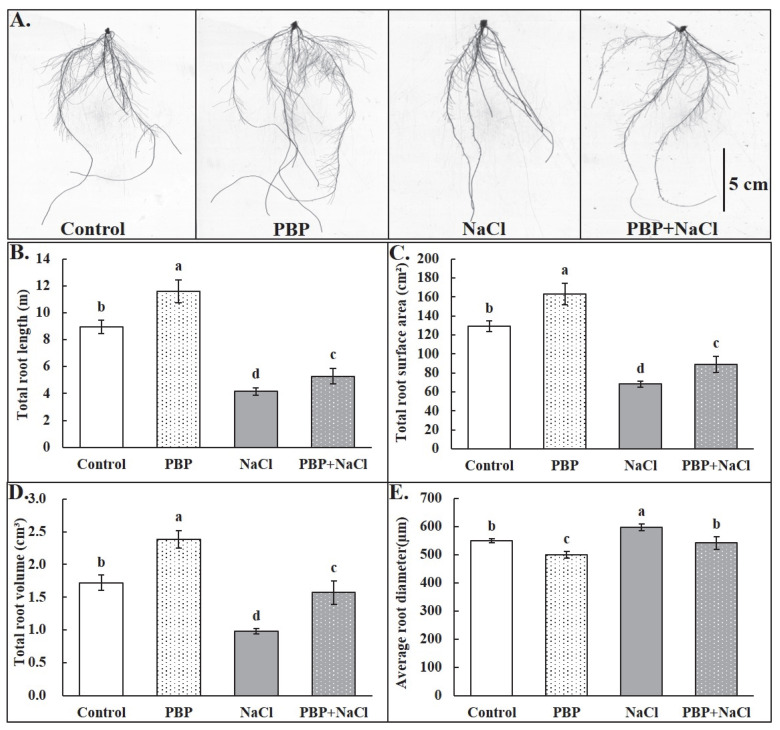
Effects of PBP on root morphology of wheat seedlings under salt stress. (**A**) Representative picture for each treatment, (**B**) total root length (TRL), (**C**) total root surface area (TRSA), (**D**) total root volume (TRV), and (**E**) average root diameter (ARD). Bars are the standard deviations (SD) of three independent replicates (*n* = 3). Error bars labeled with different letters indicate significant differences at *p* < 0.05 between treatments according to Duncan’s test.

**Figure 3 plants-14-02968-f003:**
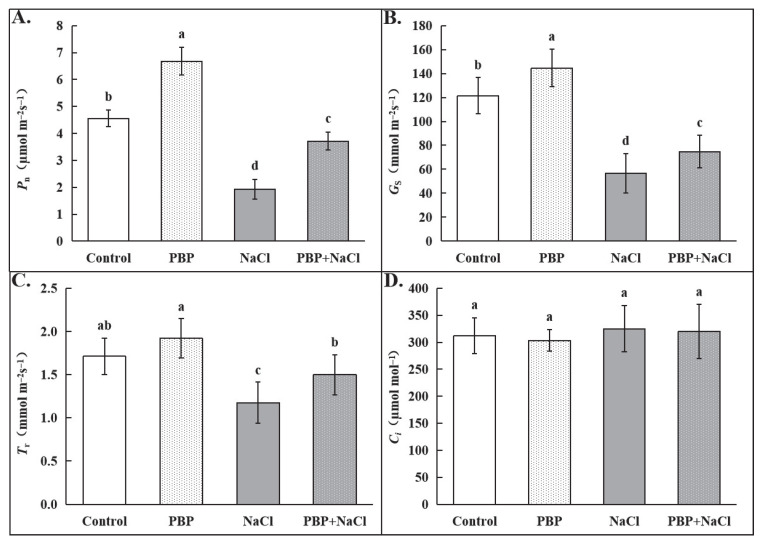
Effects of PBP on gas exchange parameters of the fully expanded leaves in wheat seedlings under salt stress. (**A**) Net photosynthetic rate (*Pn*), (**B**) stomatal conductance (*Gs*), (**C**) transpiration rate (*Tr*), and (**D**) intercellular CO_2_ concentration (*Ci*). Bars are the standard deviations (SDs) of three independent replicates (*n* = 3). Error bars labeled with different letters indicate significant differences at *p* < 0.05 between treatments according to Duncan’s test.

**Figure 4 plants-14-02968-f004:**
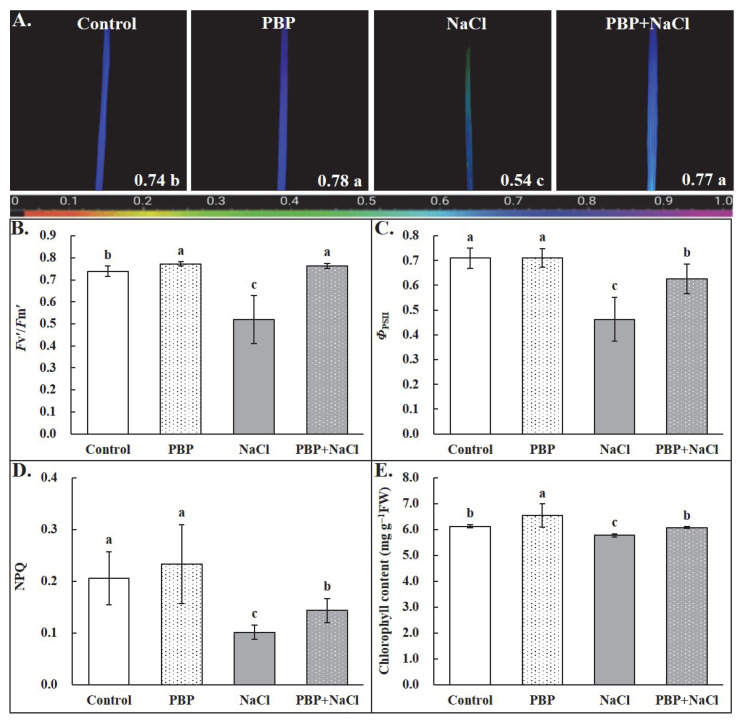
Effects of PBP on chlorophyll fluorescence parameters and total chlorophyll content of the fully expanded leaves in wheat seedlings under salt stress. (**A**) The maximal photochemical efficiency of photosystem II (PSII) (*Fv*/*Fm*). The false color code depicted at the bottom of the image ranges from 0 (black) to 1 (purple). The *Fv*/*Fm* values are depicted at the bottom of each image, (**B**) the photochemical activity of PSII (*Fv*′/*Fm*′), (**C**) the quantum yield of PSII (*Φ*_PSII_), (**D**) the nonphotochemical quenching (NPQ), and (**E**) the total chlorophyll content (Cc) expressed in mg g^−1^ fresh weight. Bars are the standard deviations (SDs) of three independent replicates (*n* = 3). Error bars labeled with different letters indicate significant differences at *p* < 0.05 between treatments according to Duncan’s test.

**Figure 5 plants-14-02968-f005:**
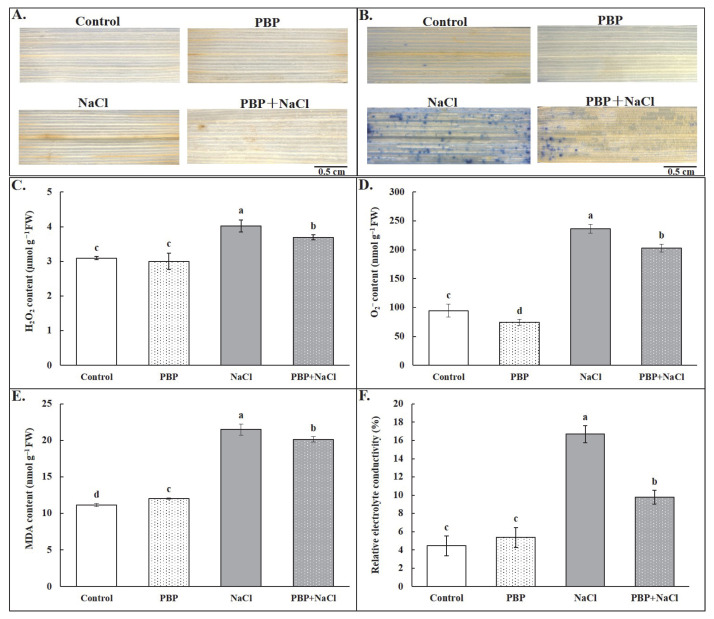
Effects of PBP on ROS accumulation and membrane damage of the fully expanded leaves in wheat seedlings under salt stress. (**A**) DAB staining reaction reflecting H_2_O_2_ accumulation, (**B**) NBT staining reaction reflecting O_2_^−^ accumulation, (**C**) H_2_O_2_ content, (**D**) O_2_^−^ content, (**E**) MDA content, and (**F**) relative electrolyte conductivity (REC). Bars are the standard deviations (SDs) of three independent replicates (*n* = 3). Error bars labeled with different letters indicate significant differences at *p* < 0.05 between treatments according to Duncan’s test. Reduced ROS accumulation and alleviated membrane damage of wheat seedlings under salt stress.

**Figure 6 plants-14-02968-f006:**
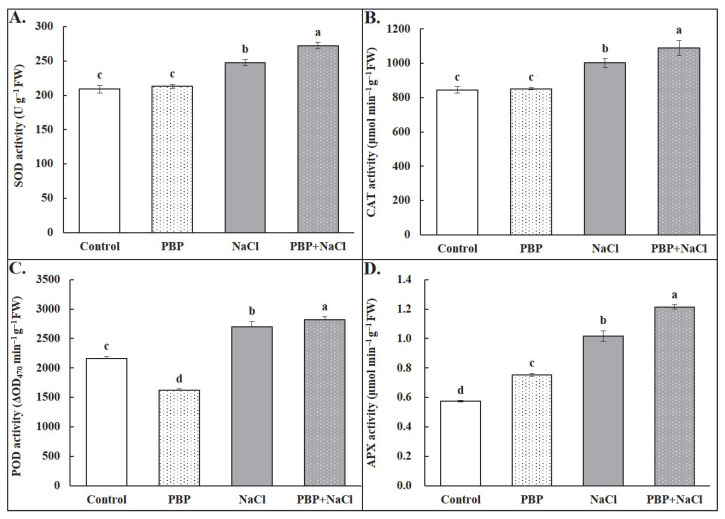
Effects of PBP on the activities of the antioxidant enzymes of the fully expanded leaves in wheat seedlings under salt stress. (**A**) Superoxide dismutase (SOD), (**B**) catalase (CAT), (**C**) peroxidase (POD), and (**D**) ascorbate peroxidase (APX). Bars are the standard deviations (SDs) of three independent replicates (*n* = 3). Error bars labeled with different letters indicate significant differences at *p* < 0.05 between treatments according to Duncan’s test.

**Figure 7 plants-14-02968-f007:**
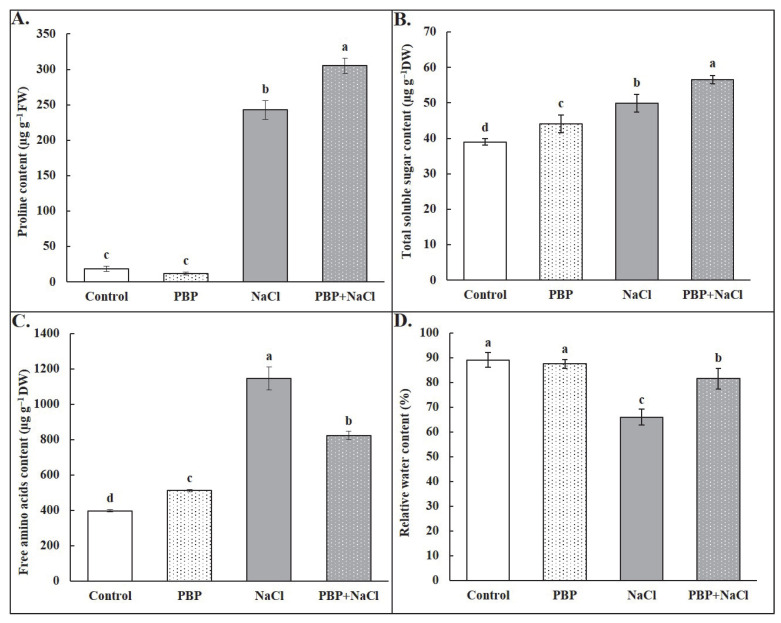
Effects of PBP on contents of osmotic regulatory substance and relative water of the fully expanded leaves in wheat seedlings under salt stress. (**A**) Proline (Pro), (**B**) total soluble sugar (TSS), (**C**) free amino acids (FAA), and (**D**) relative water content (RWC). Bars are the standard deviations (SDs) of three independent replicates (*n* = 3). Error bars labeled with different letters indicate significant differences at *p* < 0.05 between treatments according to Duncan’s test.

**Figure 8 plants-14-02968-f008:**
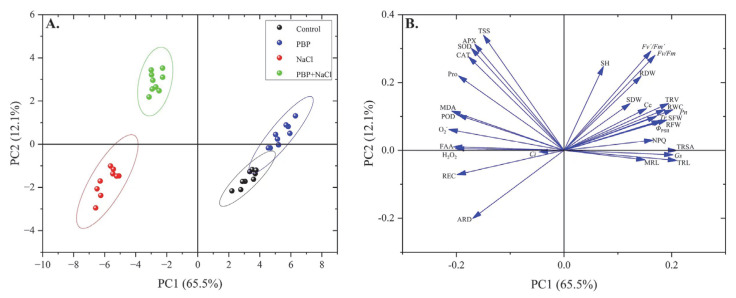
Results of principal component analysis (PCA). (**A**) Statistical analysis score diagram for the four treatments based on all parameters. (**B**) Statistical analysis score diagram of detailed parameters. Direction represents the correlation between features and length represents the strength of correlation.

**Figure 9 plants-14-02968-f009:**
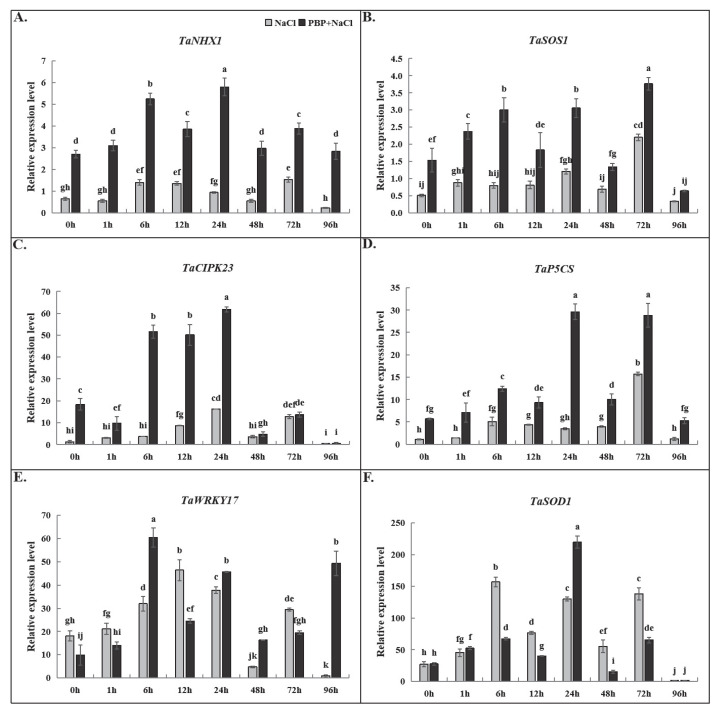
Effects of PBP on the expression of stress response genes in wheat under salt stress. (**A**) *TaNHX1*, (**B**) *TaSOS1*, (**C**) *TaCIPK23*, (**D**) *TaP5CS*, (**E**) *TaWRKY17*, and (**F**) *TaSOD1*. Bars are the standard deviations (SDs) of three independent replicates (*n* = 3). Error bars labeled with different letters indicate significant differences at *p* < 0.05 between treatments according to Duncan’s test.

## Data Availability

The original contributions presented in this study are included in the article/Appendix A. Further inquiries can be directed to the corresponding authors.

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
