# Peer review of "Priming with Porcine Blood Polypeptide Enhances Salt Tolerance in Wheat Seedlings"

_plants, 2025, doi:10.3390/plants14192968_

Round 1

Reviewer 1 Report

Comments and Suggestions for Authors

the manuscript by Shen et al presents a well-structured and detailed physiological study examining the effects of exogenous porcine blood polypeptide (PBP) on wheat (Triticum aestivum) under salt stress conditions. The topic is timely and relevant, addressing a pressing issue in global agriculture like soil salinization and proposes an innovative and underexplored solution. even if the overall study is strong, some revisions are needed.

  • please, include more details about the composition and extraction of PBP—was the preparation standardized across replicates
  • please, clarify if you have verified possible microbial contamination from PBP application
  • in several cases, some marginal statistical differences are mentioned. Please, justify also the biological relevance
  • Overall in the manuscript, terms like “new idea” should be more cautiously phrased. Moreover, explicitly discuss practical implications for field conditions
Comments on the Quality of English Language

to my  opinion the quality of English is sufficient

Author Response

the manuscript by Shen et al presents a well-structured and detailed physiological study examining the effects of exogenous porcine blood polypeptide (PBP) on wheat (Triticum aestivum) under salt stress conditions. The topic is timely and relevant, addressing a pressing issue in global agriculture like soil salinization and proposes an innovative and underexplored solution. even if the overall study is strong, some revisions are needed.

Thank you very much for your approbation and suggestions to our article.

  1. please, include more details about the composition and extraction of PBP—was the preparation standardized across replicates

We have provided a description of PBP extraction and composition as detailed as possible. However, the further specific details are kept confidential by the company. If someone want to repeat or conduct similar experiments, they can purchase this product from this company.

  1. please, clarify if you have verified possible microbial contamination from PBP application

We indeed didn’t verify if there was microbial contamination from PBP application. However, we think it may be not so important, as this research was conducted just to investigate whether exogenous PBP could alleviate the damage caused by salt stress to wheat. Of course, there might be some microorganism caused by PBP application. Besides, we think that these microorganisms might also contain some beneficial species.

  1. in several cases, some marginal statistical differences are mentioned. Please, justify also the biological relevance

Thanks for your suggestions. We have deleted the descriptions related to non-significant differences to make paper more concise and clear.

  1. Overall in the manuscript, terms like “new idea” should be more cautiously phrased. Moreover, explicitly discuss practical implications for field conditions

Thank you very much for your suggestions and reminders on cautious language. We have revised throughout the manuscript. Besides, we added a discussion on the deficiency of lacking field trials. 

Reviewer 2 Report

Comments and Suggestions for Authors

Author Response

The authors investigated the alleviating effects and physiological mechanisms of porcine blood peptide (PBP) pretreatment on wheat seedlings under salt stress, demonstrating both practical significance and innovative value in the research. The study presents a relatively comprehensive framework. However, there still are some concerns that need to be addressed.

Thank you very much for your approbation and suggestions to our article.

  1. The introduction section contains numerous “Error! Reference source not found”! These need to be thoroughly revised.

We sincerely apologize for such an error, which might be caused by not removing the EndNote formatting. We have revised thoroughly.

  1. Add scale bars to all the Figures 1A&B, 2A, 6A&B.

We apologize for our negligence and thanks a lot for your careful suggestions. We have added scale bars to these figures.

  1. Make sure that the full names of all abbreviations are indicated for the first time they appear.

Thanks a lot for your careful reminder. We have conducted a thorough review of the entire manuscript.

  1. Why were drought-resistant and salt-tolerant wheat varieties chosen? Why weren't salt-sensitive wheat varieties selected?

As salt tolerant varieties are often applied in saline-alkali fields, we just try to make the experiment as practical as possible.

  1. The authors mentioned that “ccording to our preliminary experiments (five concentrations: 0.5, 1, 2, 4 and 8g/L porcine blood polypeptide were compared, and detailed data were not shown), the effect of 4g/L porcine blood polypeptide is the most obvious.” Then, what are the effects of the other concentrations of PBP on the growth and development of wheat, promoting or inhibiting? Using a salt treatment with 250 mmol/L NaCl, is the concentration too high?

In general, each concentration of PBP could alleviate salt stress to some extent. And we have added the detailed data in the form of an supplementary figure in the revised manuscript.

As “Qingmai11” is a wheat variety with strong salt tolerance, a slightly higher salt stress concentration was set when we designed the experiment.

  1. The Materials and Methods section mentioned the use of various kits (Suzhou Greast Biotechnology Co., Ltd.) for the determination of physiological and biochemical indicators. It is necessary to supplement the item number of the key physiological indicator determination kits and the main detection principles.

Thanks for your suggestions. Item numbers of the determination kits and the main detection principles have been included in the revised manuscript.

  1. The research mainly focused on the phenotypic responses of physiological and biochemical indicators, and based on this, it was inferred that PBP enhances salt tolerance by strengthening the antioxidant system and osmotic regulation system. However, there was a lack of exploration of the potential molecular mechanisms. In the discussion section, the possible action targets and signaling pathways of PBP can be deeply explored, and more detailed speculations can be made in combination with existing literatures.

Thanks a lot for your constructive suggestions. On the one hand, we further conducted qRT-PCR analysis for several key genes involved in salt stress regulatory pathways. On the other hand, we conducted a preliminary discussion on the possible molecular mechanism of PBP.

  1. When discussing the "double-edged sword" effect of ROS, the data in the article only showed that PBP treatment reduced the level of ROS (Figure 6), but did not provide direct evidence of "moderate induction of ROS". This speculation lacks data support. It can be expressed as "PBP treatment effectively reduces the excessive accumulation of ROS induced by salt stress", while the speculation 2 about "moderate ROS as a signal" requires more caution, or it should clearly state that this is an assumption based on literatures and needs to be verified through future experiments.

Thanks a lot for your constructive suggestions. We have carefully reviewed and revised the relevant expressions based on your suggestions in the revised manuscript.

  1. Simplify or adjust the discussion in Section 3.5 regarding endogenous peptides and the CLE gene, making it more closely to explain the possible action modes of exogenous PBP. For example, could PBP possibly simulate or regulate endogenous peptide signals? Or should it be clearly listed as a future exploration direction of the main conclusion of this study?

Thanks a lot for your constructive suggestions again. These suggestions are absolutely wonderful. We have rewrite this part in the revised manuscript.

Reviewer 3 Report

Comments and Suggestions for Authors

The following are suggestions for improving this article:

The topic has potential application value, the experimental design is concise, and the data are relatively complete. However, there is still considerable room for improvement in terms of innovation, technical details, and interpretation of results (To facilitate the review process, please clearly mark the revised parts on each line).

  1. The abstract needs to highlight which indicators are important for the alleviation of salt stress by Porcine blood polypeptide.  
  2. The reference issues are too significant. "Error! Reference source not found?" It is recommended that the editor carefully pre-review before sending it to us. Such problems are not in compliance with the standards.  
  3. On Line 41, regarding the impact of salt stress on crop yield, it is suggested to cite this article(https://doi.org/10.1016/j.fcr.2025.109747)
  4. Lines 69–79 should be considered for deletion, as the physiology discussed here is not closely related to this article.  
  5. There is a problem with the transition between Lines 79 and 80. It is suggested to consider how to introduce Porcine blood polypeptide more appropriately.  
  6. Experimental design: The timing and frequency of foliar application: The application was only done twice at the four-leaf stage. It is necessary to explain why this stage was chosen and discuss the sustained effects in later stages.  
  7. The composition of the polypeptide is not clear: Please supplement the molecular weight distribution of PBP (SDS-PAGE or MALDI-TOF) and the ratio of free amino acids to small peptides.  
  8. The spraying tools, droplet size, and spreader are missing, which may affect foliar absorption. Please supplement the methodological details.  
  9. The root morphology scanning images have low resolution. A scale and magnification should be provided.  
  10. There are multiple grammatical and spelling errors in the English writing (e.g., “wherease” → “whereas”). Language polishing is needed.  
  11. The results and discussion are repetitively described. It is suggested to streamline the results section, retaining only the key significant differences, and focus the discussion on mechanisms and unknown issues.  
  12. The discussion section has significant issues. It is recommended to delve deeper into the discussion. Currently, most of it is just a restatement of the results.
Comments on the Quality of English Language

it need to improve the quality of English language. It is possible to consider giving a revision period of more than two weeks

Author Response

The topic has potential application value, the experimental design is concise, and the data are relatively complete. However, there is still considerable room for improvement in terms of innovation, technical details, and interpretation of results (To facilitate the review process, please clearly mark the revised parts on each line).

Thank you very much for your approbation and suggestions to our article.

  1. The abstract needs to highlight which indicators are important for the alleviation of salt stress by Porcine blood polypeptide.

Thanks a lot for your suggestions. We have clearly indicated that antioxidant and osmoregulatory systems play important roles in alleviating salt stress by PBP through PCA in the revised manuscript.

  1. The reference issues are too significant. "Error! Reference source not found?" It is recommended that the editor carefully pre-review before sending it to us. Such problems are not in compliance with the standards.

We sincerely apologize for such an error, which might be caused by not removing the EndNote formatting. We have revised thoroughly.

  1. On Line 41, regarding the impact of salt stress on crop yield, it is suggested to cite this article(https://doi.org/10.1016/j.fcr.2025.109747)

Thanks for your recommendation. It has been cited in the revised manuscript.

  1. Lines 69–79 should be considered for deletion, as the physiology discussed here is not closely related to this article.  

Thanks for your suggestions to the introduction. We have deleted this paragraph in the revised manuscript.

  1. There is a problem with the transition between Lines 79 and 80. It is suggested to consider how to introduce Porcine blood polypeptide more appropriately.  

Thanks for your suggestions. We have reduced this part. Please refer to the revised manuscript for details.

  1. Experimental design: The timing and frequency of foliar application: The application was only done twice at the four-leaf stage. It is necessary to explain why this stage was chosen and discuss the sustained effects in later stages.

We apologize for making foolish mistake. The correct timing and frequency of foliar application in this study is that: continuously for 5 days-once a day-during the four-leaf stage, which has been corrected in the revised manuscript. The reasons for choosing four-leaf stage are as follows. The wheat seedlings of four-leaf stage possess better adversity tolerance compared to earlier stages, which could prevent wheat seedlings from dying caused by salt stress, thereby beneficial to analyze the salt tolerance differences among four treatments in this study. Moreover, it is well known that the seedling stage is a critical sensitive period for salt tolerance, and the salt tolerance differences during this stage have a significant impact on the growth and development in the later stages.

  1. The composition of the polypeptide is not clear: Please supplement the molecular weight distribution of PBP (SDS-PAGE or MALDI-TOF) and the ratio of free amino acids to small peptides.

We have provided molecular weight distribution of PBP. However, the company did not conduct the ratio analysis of free amino acids to small peptides for PBP. We are extremely sorry and very regretful.

  1. The spraying tools, droplet size, and spreader are missing, which may affect foliar absorption. Please supplement the methodological details.

The spraying tools and droplet size have been included in Materials part of the revised manuscript.

  1. The root morphology scanning images have low resolution. A scale and magnification should be provided.

A scale has been added in the revised Figure 2. When we were conducting the experiments, we thought these pictures and quantitative analysis data were sufficient to demonstrate the differences among the four treatments. Now considering your suggestions, it is indeed true that a higher resolution and magnifications would be better. However, limited days was given for the manuscript revision, so we apologized that we can’t provide figures with higher resolution, as well as magnifications.

  1. There are multiple grammatical and spelling errors in the English writing (e.g., “wherease” → “whereas”). Language polishing is needed.

We apologize for the lack of standardization in our English writing, especially the spelling errors. Language polishing has been conducted in the revised manuscript.

  1. The results and discussion are repetitively described. It is suggested to streamline the results section, retaining only the key significant differences, and focus the discussion on mechanisms and unknown issues.

Thanks a lot for your sincere suggestions. We have streamlined the results section, especially deleting the descriptions related to non-significant differences, to make paper more concise and clear. Under your reminder, we also realized that the results and discussion are indeed repetitively described to some extent. We have made significant revisions to the discussion section. Please review.

  1. The discussion section has significant issues. It is recommended to delve deeper into the discussion. Currently, most of it is just a restatement of the results.

Thanks a lot for your sincere suggestions again. We have made significant revisions to the discussion section.

Reviewer 4 Report

Comments and Suggestions for Authors

Overall, the paper is conventional but too generic, lacking innovation. The experimental focus is overly narrow—it is recommended to explore salt tolerance from multiple perspectives.

  1. Figures 1A, 1B, and 2A lack scale bars.

  2. The entire paper only discusses the effects of PBP on wheat salt tolerance through physiological indicators, lacking molecular-level experiments such as measuring the expression levels of genes in salt stress regulatory pathways via qPCR.

  3. Differences marked by different letters should be explained in the captions of each figure, rather than being mentioned only once in the Materials and Methods section.

  4. Some images (Figures 2 and 7) are misaligned in their composition, and the visual presentation needs improvement.

  5. More physiological indicators do not necessarily mean better. Since this study primarily focuses on salt stress, analyzing leaf photosynthesis should suffice—is the photosystem efficiency (PSâ…¡) in Figure 5 really necessary? Additionally, the roots are the most affected by stress, yet all physiological indicators were measured in the leaves. Even the osmotic-related physiological indicators in Figure 8 were derived from leaves. The only root-related metrics mentioned in the paper are physical parameters like length and volume, with no mention of chemical-level substance content.

Author Response

Overall, the paper is conventional but too generic, lacking innovation. The experimental focus is overly narrow—it is recommended to explore salt tolerance from multiple perspectives.

Thank you very much for your constructive suggestions to our article.

  1. Figures 1A, 1B, and 2A lack scale bars.

We apologize for our negligence and thanks a lot for your careful suggestions. We have added scale bars to these figures.

  1. The entire paper only discusses the effects of PBP on wheat salt tolerance through physiological indicators, lacking molecular-level experiments such as measuring the expression levels of genes in salt stress regulatory pathways via qPCR.

Thanks a lot for your constructive suggestions. For six classical genes involved in salt stress regulatory pathways, we conducted qRT-PCR analysis.

  1. Differences marked by different letters should be explained in the captions of each figure, rather than being mentioned only once in the Materials and Methods section.

Thanks for your suggestions. It has been explained in the captions of each figure in the revised manuscript.

  1. Some images (Figures 2 and 7) are misaligned in their composition, and the visual presentation needs improvement.

Thanks for your careful reminder to the figures. We have improved them.

  1. More physiological indicators do not necessarily mean better. Since this study primarily focuses on salt stress, analyzing leaf photosynthesis should suffice—is the photosystem efficiency (PSâ…¡) in Figure 5 really necessary? Additionally, the roots are the most affected by stress, yet all physiological indicators were measured in the leaves. Even the osmotic-related physiological indicators in Figure 8 were derived from leaves. The only root-related metrics mentioned in the paper are physical parameters like length and volume, with no mention of chemical-level substance content.

Aiming to conduct a comprehensive and systematic analysis to understand the role of PBP in alleviating salt stress in wheat at the physiological and biochemical levels, more physiological indicators were measured. Besides, to our knowledge, analysis of chlorophyll fluorescence parameters under salt stress is necessary, which can provide real-time and dynamic information about the functional status of the photosynthetic apparatus,while gas exchange parameters just directly quantify carbon assimilation that is a result-oriented reflection to some extent.

Indeed, studies on the root system are of great significance in the analysis of salt stress responses. The reasons that physiological indicators were measured focusing on the leaves in this study are as follows. Take concentrations of osmotic regulatory substance as an example. 1) The mechanism of leaf osmotic regulation is more comprehensive, involving aspects such as stomatal regulation and photosynthetic protection, thereby can better reflect the overall salt tolerance ability. 2) Leaf osmotic regulation and root osmotic regulation achieve systematic coordination through hormone signals and other means, and exert a synergistic adaptive mechanism. 3) Most physiological experiments tend to use leaves as the measurement material, as they are easier to obtain and the measurement results have higher stability.

Round 2

Reviewer 2 Report

Comments and Suggestions for Authors

The authors have addressed the questions.

Reviewer 3 Report

Comments and Suggestions for Authors

good revisions